# Preliminary Assessment of Microbial Community Structure of Wind-Tidal Flats in the Laguna Madre, Texas, USA

**DOI:** 10.3390/biology9080183

**Published:** 2020-07-22

**Authors:** I.-Shuo Huang, Lee J. Pinnell, Jeffrey W. Turner, Hussain Abdulla, Lauren Boyd, Eric W. Linton, Paul V. Zimba

**Affiliations:** 1Center for Coastal Studies, Texas A&M University-Corpus Christi, Corpus Christi, TX 78412, USA; paul.zimba@tamucc.edu; 2Virginia Institute of Marine Science, William & Mary, Gloucester Point, VA 23062, USA; 3Department of Life Sciences, Texas A&M University-Corpus Christi, Corpus Christi, TX 78412, USA; lpinnell@sheddaquarium.org (L.J.P.); jeffrey.turner@tamucc.edu (J.W.T.); 4A. Watson Armour III Center for Animal Health and Welfare, John G. Shedd Aquarium, Chicago, IL 60605, USA; 5Department of Physical and Environmental Sciences, Texas A&M University-Corpus Christi, Corpus Christi, TX 78412, USA; hussain.abdulla@tamucc.edu; 6Department of Biology, Central Michigan University, Mount Pleasant, MI 48859, USA; boyd3lm@cmich.edu (L.B.); linto1ew@cmich.edu (E.W.L.)

**Keywords:** Hypersaline lagoonal wind-tidal microbial flats, Laguna Madre, cyanobacteria, MiSeq, toxins, UPLC-Orbitrap MS

## Abstract

Aside from two samples collected nearly 50 years ago, little is known about the microbial composition of wind tidal flats in the hypersaline Laguna Madre, Texas. These mats account for ~42% of the lagoon’s area. These microbial communities were sampled at four locations that historically had mats in the Laguna Madre, including Laguna Madre Field Station (LMFS), Nighthawk Bay (NH), and two locations in Kenedy Ranch (KRN and KRS). Amplicon sequencing of 16S genes determined the presence of 51 prokaryotic phyla dominated by Bacteroidota, Chloroflexi, Cyanobacteria, Desulfobacteria, Firmicutes, Halobacteria, and Proteobacteria. The microbial community structure of NH and KR is significantly different to LMFS, in which Bacteroidota and Proteobacteria were most abundant. Twenty-three cyanobacterial taxa were identified via genomic analysis, whereas 45 cyanobacterial taxa were identified using morphological analysis, containing large filamentous forms on the surface, and smaller, motile filamentous and coccoid forms in subsurface mat layers. Sample sites were dominated by species in Oscillatoriaceae (i.e., *Lyngbya*) and Coleofasciculaceae (i.e., *Coleofasciculus*). Most cyanobacterial sequences (~35%) could not be assigned to any established taxa at the family/genus level, given the limited knowledge of hypersaline cyanobacteria. A total of 73 cyanobacterial bioactive metabolites were identified using ultra performance liquid chromatography-Orbitrap MS analysis from these commu nities. Laguna Madre seems unique compared to other sabkhas in terms of its microbiology.

## 1. Introduction

Surface biotic microlayers are essential in many habitats for soil stabilization [1,2], building behavior of aquatic animals [3], and surface aquatic biofilm formation [4]. Surface layers can be termed biofilms, biological crusts, or mats, depending on development stage [1,5]. Biological crusts have been extensively studied in desert and polar regions, and include fungal, bacterial, and algal constituents [5,6]. Although most common in coastal and hypersaline areas (Table 1), these mats can occur in freshwater hot springs [7] and alkaline lakes [8,9].

The Laguna Madre is the largest hypersaline system in the world if the international border that splits the Laguna Madre into the US portion and Mexican Laguna Tamaulipas is ignored [10]. This ecosystem is renowned for its fishery, clear water, and extensive seagrass beds. The Laguna Madre is located on the Central Flyway and serves as the winter feeding grounds for many endangered avian species [11]. Over 40% of the Laguna’s area is covered by wind tidal algal flats, based on Bureau of Economic Geology mapping efforts [12]. A unique feature of the Laguna Madre is a constriction in the middle of its length by sand—this is a coastal sabkha unlike those found in the Persian Gulf and Mediterranean Seas. Laguna Madre water is occasionally pushed by wind over this shallow area, forming brines that are mixed into deeper lagoonal water [10,12,13]. Little is known about the benthic microbial community of Laguna Madre occurring in these tidal flats [14,15]. In each case, a single algal sample was examined by a noted phycologist who identified the dominant taxa present. Collectively, the community was identified to be dominated by *Lyngbya aestuarii* and *Microcoleus chthonoplastes* (two cyanobacterial species) with occasional diatoms (e.g., *Amphora normanii* Rabenhorst, *Amphora* spp., *Nitzschia* spp.) present. Pulich and Rabalais [15] conducted nitrogen fixation experiments seasonally, finding significant “new” N resulting from nitrogen fixation in the algal mats. Mats can consist of up to three centimeters’ depth with the attachment of sand and clays to the biological consortium [16]. During the past 80 years, considerable alterations have occurred within the Laguna [10]. These include: dredging of the intercoastal waterway that increased water depth to ~4 meters in the main channel (the average depth otherwise is <1 m), opening several passes through the barrier island, removing the sill between the Laguna and Corpus Christi Bay, as well as increased nutrient loading from development, including housing and agriculture [10].

As is widely appreciated, taxonomic analysis of mat communities is nearly impossible using morphological criteria alone for cyanobacteria [2,5] and for bacteria (Table 1). This study used a polyphasic approach, including morphological, isolation/culturing, next-generation analyses, and chemical approaches, to describe the biodiversity within these complex biofilm/mat communities. There is an urgency to understand unique habitats such as these hypersaline mats, as sea-level rise and Anthropocene influences are likely to alter their habitat. The isolation and identification of these organisms may provide unique resources for natural product discovery.

## 2. Materials and Methods

### 2.1. Study Site

Sandflat algal mat samples for morphological, isolation/culturing, next-generation, and toxin analyses were collected from 4 different sample sites in the Laguna Madre during May 2018 (Figure 1). Each site contained an elevational gradient of wetting and were identified as: Kenedy Ranch (KRN and KRS; 2 transects about 1.3 km apart, near Tres Marias Island), Nighthawk Bay (NH; just north of Bird Island), and on Laguna Madre Field Station island (LMFS). Samples were collected along a gradient from wet (nearly submersed), an intermediate site with no visible water, and a dry site more shoreward in a sterile plastic container (18 cm × 32 cm × 5 cm deep) and immediately placed in a darkened cooler on ice packs until processed in the laboratory.

### 2.2. Microscopy

Structure of the mats was captured using an Olympus SZ60 stereo microscope (Olympus, Tokyo, Japan) that was equipped with an AmScope 10 MP Microscope Digital Camera, MU1000 controlled with AmScope v3.7.1 software (AmScope, Irvine, CA, USA). Species composition of the cyanobacteria were examined using either a Wild M5 inverted microscope (Wild Herrburgg, Heerbrugg, Switzerland) or a Zeiss Axio Imager A1 Microscope (Carl Zeiss Inc., Hallbergmoos, Germany). Taxa were named according to the current nomenclature [35,36,37,38,39]. Identified cyanobacterial taxa were classified as dominant (>51%), common (20%), and rare (5% or less) according to their occurrence on the surface of the mat. In some cases, identification occurred from enrichment cultures grown in f/100 media [40] at salinity 47 ppt. Subsamples of each site were viewed alive and after being preserved using 4% formalin. For SEM examination, samples were dehydrated through an ethanol gradient (25%, 50%, and 75%) to a final concentration of 100% and then transferred to a coverslip. Samples were chemically dried using hexamethyldisilazane (cat. #379212; Sigma-Aldrich, St. Louis, MO, USA), then attached to an aluminum stub using carbon adhesive tape and coated with gold/palladium using a Denton Desk II sputter coating system (Denton Vacuum, LLC, Moorestown, NJ, USA). Samples were examined using a Hitachi 3400 N-II Scanning Electron Microscope (Tokyo, Japan). Photo plates were assembled in Microsoft PowerPoint (Microsoft Cooperation, Redmond, WA, USA) and Adobe Photoshop CC (Adobe System Inc., San Jose, CA, USA).

### 2.3. Molecular Methods and Sequencing

Total DNA was extracted using a DNeasy PowerSoil Kit (QIAGEN, Hilden, Germany). The V4 region of the 16S rRNA gene was amplified using a HotStarTaq Plus Master Mix Kit (QIAGEN) and the improved 515f and 806r primer set with barcoded forward primers [41]. Amplification conditions were 94 °C for 3 minutes followed by 28 cycles of 30 seconds at 94 °C, 40 seconds at 54 °C, and 72 °C for one minute. Final elongation occurred at 72 °C for 5 minutes. Amplification was checked for success visually using gel electrophoresis, and samples were pooled in equal proportions based on their molecular weight and DNA concentrations. Pooled PCR products were then purified using calibrated Ampure XP beads (Beckman Coulter, Indianapolis, IN, USA), and the resulting pooled library was sequenced on an Illumina MiSeq instrument using paired-end chemistry (2 × 150 bp) at Molecular Research LP (Shallowater, TX, USA). Raw sequence reads were processed using a combination of QIIME v1.9 and QIIME2 version 2020.2 [42]. Barcodes were extracted from the paired-end reads using the ‘extract_barcodes.py’ tool within QIIME v1.9, and then imported into QIIME2 for the remainder of the analysis. Sequence reads were demultiplexed and then denoised using DADA2, using trim lengths of 242 bp and 233 bp on the forward and reverse read, respectively. In addition, DADA2 improves sequences quality by removing chimeric sequences, and merges paired-end reads. Principal co-ordinate analysis of weighted UniFrac values was also carried out using the q2-diversity plugin. Taxonomy was assigned using a Naïve Bayes classifier trained on the SILVA release 138 operational taxonomic units (OTUs) database at 99% criterion, where sequences had been trimmed to include only the 250 bases from the 16S region that were sequenced. Taxonomic bar plots were generated using ggplot2 package (ver. 3.3.0) and biodiversity was analyzed using Bray Curtis dissimilarity index calculated from the vegan package (ver. 2.5.6) in RStudio program (R ver. 4.0.0) to visualize taxonomic differences between samples.

### 2.4. Mat Cyanotoxin Analysis

Presence of toxins was determined using an ultra performance liquid chromatography (UPLC) Thermo Scientific Orbitrap Fusion Tribrid mass spectrometer (ThermoScientific Waltham, MA, USA). A library of known cyanobacterial metabolites was used for confirmation of retention time [43]. Toxins were extracted after a 4-hour period using acidified acetonitrile [44] at 4 °C, then clarified using 0.2 µm filters prior to analysis. Samples were loaded onto a C18 column (3 × 100 mm, 3 µm particle size, Phenomenex Corp., Torrance, CA, USA), with formic-acid-added ultrapure water and acetonitrile as mobile phases. As standards for most toxins were not available, toxin concentration was not determined. Toxin confirmation was achieved through the alignment of the retention time, the difference in predicted and detected masses (<3 ppm), the peak abundance (>10^4^ unit), and the isotope patterns.

## 3. Results

### 3.1. Microscopic Based Taxonomic Structure of the Mat

Mats from all locations were composed of layered organic and inorganic strata largely occupied by cyanobacteria, with infrequent pennate diatoms, euglenoids, and flagellate algae present in the upper 1 cm. Cyanobacteria formed a thin surface layer 0.1–0.5 mm with inorganic material associated with the sheath of at least some taxa. During collections, the only animals found were dipteran flies in mats and annelids occasionally in the submerged mats at LMFS and NB.

Mats from submersed and dry regions contained different algal communities (Figure 2A–F and Figure 3A–AB). Submerged mats were less developed in terms of surface coverage and were less thick (Figure 2A,B). Based on light microscopy, the species richness of cyanobacteria in the microbial mats from the Laguna Madre was high, with 45 species/unique morphotypes identified (Table 2). Of these, 27 taxa were filamentous forms (62% of the flora) from seven different orders. Synechococcales, Spirulinales, Oscillatoriales, Nostocales were dominant numerically. Other taxa were identified from the Pleurocapsales, Chroococcales, and rarely Chroococcidiopsidales. Species richness of the individual samples was ~10 species always with at least one dominant taxon. *Coleofasiculus* spp. accounted for ~70% of the taxa present in the submerged sites (Figure 3E), with “*Lyngbya*-like*”* filaments commonly found. (Figure 2C, 3N,O). Collectively these taxa accounted for ca. 95% of the taxa on the sediment surface but were rarely found buried within the mat. *Coleofasciculus chthonoplastes* found in the Laguna Madre (Figure 3E) was similar to the type species *C. chthonoplastes* WW7 isolated from the wind flats of the Baltic Sea, Germany [45]. Morphologically the “*Lyngbya*” sp. from the Laguna Madre was similar to *Lyngbya aestuarii* Liebman ex Gomont, except our material lacked the calyptra found in this taxon (Figure 3O). A brown sheath covering *Lyngbya* was evident (Figure 3N). Species of several cyanobacteria genera (i.e., *Geitlerinema, Oscillatoria*, *Pseudanabaena*, and *Spirulina*) were commonly found in the microbial mats from the Laguna Madre (Figure 3Q,V,X). The taxon *Leibleinia subtilis* (Figure 3M) was rarely encountered, growing coiled on “*Lyngbya*” sp. A newly described taxon, *Perforafilum tunnelli*, with a tapered end and boring ability, was rarely found in the mats (Figure 3R). Most coccoid cyanobacteria were rarely found in the Laguna Madre microbial mat, except *Aphanocapsa*, *Bacularia*, and *Synechococcus* (Figure 3B,AA). Unique multicelluar-colonial *Chroococcus*, *Cyanosarcina*, *Eucapsis*, *Gloeocapsa*, and *Gloeocapsosis* were identified in the Laguna Madre mats with rare abundance (Figure 3D,G–J). A rare pleurocapsalean taxon *Pleurocapsa* sp. (Figure 3T) was found on shell fragments of the bivalve *Anomalocardia auberiana. Pleurocapsa* sp. and the rarely encountered coccoid species *Chroococcus* cf. *pulcherrimus* (Figure 3F) were enveloped by a brown sheath. A poorly know taxon, *Johannesbaptistia* sp., was found intermixed with other cyanobacteria in the mat (Figure 3K). This taxon’s morphometric measurements and morphology differed from values reported for the type species, *Johannesbaptistia pellucida*. Cyanobacterial genera known from the literature to be bioactive metabolite producers were present, including *Lyngbya*, *Nodosilinea*, *Oscillatoria*, *Perforafilum*, *Phormidium*, *Pleurocapsa*, *Pseudanabaena*, *Schizothrix*, *Spirulina*, and *Synechococcus* (Figure 3N–T,V–X,AA).

### 3.2. Molecular and Phylogenetic Analysis

The complete OTU table is summarized in the Appendix A. According to principal coordinate ordinations (Appendix A), microbial communities of KRS and KRN sample sites (all conditions) tended to group together; however, that trend was not significant. The dry and medium sites of NH grouped with KR sample sites, supporting the idea that dry sites within LMFS and NH are taxonomically similar to wet sites in KR. The microbial community from the sand flats of the Laguna Madre were dominated by members of the phyla Proteobacteria and Bacteroidota, which collectively accounted for 20–60% of OTU relative abundance (Figure 4). Bray–Curtis dissimilarity index showed that the LMFS-dry site had a unique microbial composition dominated by Firmicutes and Proteobacteria, whereas the microbial compositions of the other three dry sites were similar. All sites in KR shared similar microbial composition at the phyla level to the two NH sites, i.e., NH-medium and NH-dry. KRN-wet sites were dominated by Chloroflexi, followed by Proteobacteria and Bacteroidota. According to MiSeq analysis, 50 microbial phyla were identified from KR mats. Members of the phylum Cyanobacteria represented less than 15% of the total relative abundance of the microbial community from all sample sites and all conditions (Figure 4); the KRS-medium wetness site had the largest Cyanobacteria abundance, which was double the next closest site (KRN-dry).

Eighteen Bacteriodota families were identified in the Laguna Madre microbial community. Balneolaceae, Cyclobacteriaceae, Rhodothermaceae, and Saprospiraceae were the four most abundant families in the KR microbial mats, accounting for 45%~95% (mean = 72%) of the Bacteriodota OTUs (Appendix A). Species of Rhodothermaceae were less abundant in LMFS sites compared to the three other locations. Amoebophilaceae and Marinilaniliaceae were only found in LMFS and NH sites, whereas Salinivirgaceae was only present in NH sites. The Chloroflexi were dominated by unnamed species, with four families identified including the Anaerolineaceae, Ardenticatenaceae, Caldilineaceae, and Chloroflexaceae (Appendix A). The KRN-wet site was the only one that contained all four Chloroflexi families. Sixteen Desulfobacterota families were identified from the Laguna Madre. Four families were commonly present in all sites, including Bradymonadaceae, Desulfocapsaceae, Desulfococcaceae, and Desulfosarcinaceae (Appendix A). Another family, the Desulfovibrionaceae, was found in all sites but LMFS-wet; further, the abundance of this particular family in three LMFS sites was lower than other sites. Species of Desulfurivibtionaceae and Geothermobacteraceae were only found at the KRN sites. According to Bray–Curtis indices, it is clear that the composition of Desulfobacterota in LMFS and NH was different than KR sites. Firmicutes accounted for 25% of microbial abundance at LMFS-dry (Appendix A), whereas at LMFS-wet they were not present. Fifteen families in Firmicutes were identified, with Bacillaceae found in all sites but NH-dry. Fusibacteraceae were only found in LMFS and NH sites. Lachnospiraceae were only found at the NH-wet site. Thermotaleaceae were only found in KRN sites, i.e., KRN-medium and -dry locations. Five Halobacterota families were identified in the Laguna Madre mats, with Haloferacaceae and Halomicrobiaceae being the dominant families (Appendix A). KRN-dry and KRS-wet had all five families present, whereas both LMFS and NH wet sites had no occurrence of Halobacterota. Halobacteriaceae and Halococcaceae were only found in KR sites. A total of 58 proteobacterial families were found in the Laguna Madre mats (Appendix A). Chromatiaceae and Rhodobacteraceae were commonly found in all sites, accounting for 10% to 50% of the OTUs, averaging 28% (Appendix A). Fodinicurvataceae had high abundance at all sites in NH and KR, other than the three sites in LMFS.

Species in the cyanobacterial families Coleofasiculaceae and Oscillatoriaceae were the most abundant in the mats, accounting for 25–85% (average 46%) of the cyanobacteria OTUs (Figure 5). The dry site at LMFS was the only site that did not contain species of Oscillatoriaceae. Unlike the Oscillatoriaceae, species in the Coleofasiculaceae were mostly found in dry areas. Coleofasiculaceae were either not found or present in low OUT abundance (<20%) in samples from all four wet sites across the Laguna Madre, i.e., KRN-wet, NH-wet, LMFS-medium and -wet. Leptolyngbyaceae were found in 10 sites, but not at KRN-wet and LFMS-dry. Microcoleaceae were only found in rare abundance at the other three low moisture sites, including NH-medium, KRN-medium, and KRS-dry. Chroococcaceae and Spirulinaceae were only found in LFMS and NH, whereas Merismopediaceae were only found in KR sites.

Total species richness of cyanobacteria, using the SILVA automatic classification, was 23 known taxa (Figure 6) within nine families (Appendix A). Nine of the 23 taxa were not identified by light microscopy including species in the genera *Arthrospira, Dactylococcopsis*, *Gloeothece*, *Halospirulina*, *Hydrocoleum*, *Leptolyngbya, Nodosilinea, Roseofilum*, and *Symploca*. According to OTU analysis, *Lyngby* sp. PCC-7419 (30% of the total), *Coleofasciculus* sp. PCC-7420 (20%), and *Phormidium* sp. (8%) were the most abundant taxa in the mats (Figure 6). Overall, ~35% of OTUs were not matched to known species in the SILVA database. The medium and dry sites of KRN had the highest species richness of *Lyngbya* (Appendix A), supporting our idea that “*Lyngbya*” spp. are best suited for growth/survival in these dry environments and that multiple cryptic species are present (Table 2). All sites in LMFS, as well as wet and medium sites of NH, did not contain any “*Lyngbya*” spp. sequences. *Coleofasciculus* was most abundant in LMFS-dry, which differs from our microscopic data, where *Coleofasciculus* spp. were dominant only in the wet sites at LMFS and NH. Generally, filamentous and coccoid cyanobacteria co-occurred in the sand flat mats of the Laguna Madre, including hypersaline and thermo-tolerant forms, e.g., *Halospirulina* and *Cyanocohniella*.

### 3.3. Cyanobacterial Bioactive Metabolite Analysis

A total of 57 toxins were identified from the mat samples (Table 3). The most abundant class of toxins found are the aeruginosins (14), with 10 unique microcystins, 7 microcycloamides, and 6 cyanopeptolins identified. One lynbelloside and two aeruginosins had high area counts (exceeding 10,000). Cyanobacterial genera that are reported in the literature to produce six of 57 bioactive metabolites were identified in this study, whereas the cyanobacterial family reported as the original source of 28% metabolites were identified. About 50% of the metabolites present were originally isolated from *Microcystis* in the literature, of which eight associated genera (e.g., *Aphanothece* and *Chroococcus*) were identified using morphological and genomic analyses. Four bioactive metabolites found in this study were originally isolated from marine sponge *Theonella* sp. and seahare *Dolabella* sp., indicating the source should be their dietary cyanobacteria.

## 4. Discussion

Climate change is affecting numerous ecosystems worldwide, including widespread alteration of tidal flats [46]. Mats positioned between the ocean and coastlines are impacted by land hardening, population growth/nutrification, as well as sea level rise. It is estimated that 16% of these areas have been destroyed in the past 32 years [46,47]. Changes of centimeters in elevation (sea-level rise) inundate mats more frequently, thereby allowing less hardy species to proliferate. Unique habitats such as the Laguna Madre are susceptible to changing environmental conditions, because of the selective unique conditions that helped organisms adapt to this unique sabkha. Alterations of salinity (from levels up to 100 practical salinity unit to near seawater), increased wetting frequency of the mats due to channelization of the Intercoastal Waterway and opening more passes to the Gulf of Mexico, and alteration/development of land collectively threaten to alter these mats in terms of species composition and biomass present [10]. Increases in water level or inundation frequency will affect mat composition, as these new disturbances (in the form of salinity decreases) have been shown to alter composition of saline pond communities of San Salvador Island, Bahamas [29]. Increased biodiversity of cyanobacteria was associated with the near four-fold decrease in salinity after hurricane passage. Climate changes may alter species composition or physiological activity of the microbial consortia. Exopolysaccharide secretion are typically required for initiation of adhesion of bacteria [4,48] and cyanobacteria [49] to the inorganic soil/sediments. This adhesive layer is critical for mat stability during desiccation periods, to develop more diverse composition [5,49], and for differential metabolite formation in pathogenic bacteria [4]. The dominant taxon in the Laguna Madre, *Lyngbya*, can produce copious amounts of extracellular polymeric substances [50] and so changes in environmental conditions may threaten its current architecture, functionality, and ecosystem role.

Mat dominance by bacteria, particularly Proteobacteria and Bacteroidetes, have been observed in other hypersaline communities (Table 1). Vogt et al. [23] reported up to 60% dominance of Bacteroidetes in the hypersaline microbial communities of the Gulf of Oman, with similar results from the surface of the microbial mats from the hypersaline Guerrero Negro, Baja California, Mexico [27,28]. Both Proteobacteria and Bacteroidetes possess diverse nitrogen metabolism and sulfur reduction [51,52,53,54,55]. Other than Bacteroidetes and Proteobacteria, species in Chloroflexi, Desulfobacteria, Firmicutes, and archaea-Halobacterota were also commonly present in the Laguna Madre mats. Chloroflexi have been reported from thermal environments and are known for their anaerobic decomposing ability [56,57], whereas Halobacterota are well-known for their occurrence in hypersaline environments and utilization of cellulose and chitin as nutrient sources [58,59]. Unlike other hypersaline sites, Laguna Madre has abundant oil and natural gas deposits. The drilling process of harvesting these natural sources may have resulted in the richness of sulfur-containing substances, which support the success of species of Desulfobacteria. Alternatively, the seasonal deposition of seagrass wrack provides an anaerobic habitat suitable for growth of these taxa. The occurrence of Firmicutes was ~25% at LMFS-dry while it was almost absent from other sampled sites. This particular site was nearly always (~90% of time) exposed to the air, allowing the community to be largely affected by the low-moisture conditions and solar radiation. The ultra-structure of Firmicutes, preventing damage from solar radiation [60] and the loss of intracellular moisture [61], may explain the abundance of this phylum at LMFS-dry compared to other phyla. Further, the presence of a rare phylum, Planctomycetota, suggests the potential for anaerobic ammonium oxidation to occur in the Laguna Madre microbial mats [62]. The microbial composition revealed by next-generation sequencing suggested the Laguna Madre mats are unique compared to other locations.

Taxonomic cyanobacterial composition of Laguna Madre mats was similar to other tropic/sub-tropical hypersaline systems (Table 2) with a total of 55 taxa found using either light microscopy or Next-generation 16S rRNA analyses. Of these, 13 taxa were found using both approaches, with the dominance of *Lyngbya* (~30% overall) and *Coleofasiculus* (~18%). High abundance of *Coleofasciculus* spp. at the medium and dry sites in the Laguna Madre based on SILVA automatic classification data suggested the preference of this genus regarding wetness (Appendix A). This occurrence of *Coleofasciculus* is opposite to the finding in Vogt et al. [23], which reported *Coleofasciculus* prefer lower tidal sites (high wetness conditions). Both morphological observation and SILVA automatic classification revealed several species of *Coleofasciculus*, but none of these analyses could resolve species level taxonomy. Unlike *Coleofasciculus*, *Lyngbya* was rarely found in NH and LMFS sites (except NH-dry), but was commonly occupied in the samples from KR, especially KRN-medium and -dry sites, as *Lyngbya* were present in the medium to dry sample sites from other studies [23]. This distribution could be a result of (1) the favorable conditions, and (2) the structure of *Lyngbya*. Both NH and LMFS sites were exposed to the direct current in the Laguna Madre, whereas two KR locations were at a divergence from the main Laguna Madre stream (Figure 1). Compared to KRS, which was along the shore of KR, KRN was at an angle that could decelerate the water flow. With the significant dominance of *Lyngbya* at KRN, it is likely this particular taxon favors a slow/static environment. An alternative explanation is that heavily sheathed “*Lyngbya*” spp. had poor extraction of DNA. Cyanobacterial taxa such as *Cyanocohniella* and *Oscillatoria* have been identified in other hypersaline and thermal environments [63,64] and were present in this study (Figure 3F,Q). Many of the taxa do not fit within described taxa for either the dominant (*Lyngbya* and *Coleofasiculus*, for example) or rarer taxa (*Perforafilum tunnelli*) as seen in other analysis of benthic saline cyanobacteria [29,50]. We feel this is a result of limited analysis of benthic hypersaline communities, as recent taxonomic revisions have resulted in several new genera and species such as *Dapis* being separated from *Lyngbya* [65], and *Toxifilum,* a cryptic taxon from the hypersaline Corpus Christi Bay, serving to establish a new family (the Laspinaceae) [66]. According to Vogt et al. [23], coccoid cyanobacteria such as *Euhalothece* sp. and species in Chroococcales had dominated the upper tidal site (high dryness conditions). There was no general trend in the occurrence of coccoid cyanobacteria in this study due to the dominance of filamentous cyanobacteria in all sites.

Cyanobacterial genera that have been reported as bioactive metabolite producers were present in the Laguna Madre microbial mats, including *Nodosilinea*, *Lyngbya*, *Oscillatoria*, *Perforafilum*, *Phormidium*, *Pleurocapsa*, *Pseudanabaena*, *Schizothrix*, *Spirulina*, and *Synechococcus* (Table 3) [43]. Given the widespread production of cyanobacterial bioactive metabolites (Table 3), it is tempting to suggest that these metabolites may serve as a deterrent for growth of animals and microbes, but it is also likely that the lack of water for animal survival may better explain the absence of grazing organisms. Identified bioactive compounds that had high peak abundance (>10^6^) have documented protein phosphatase inhibition and cytotoxic activity (Table 3) [43]. These mats appear to be a rich, diverse source of numerous poorly known toxins [67]. Our culturing and analysis approach, allowing the systematic assignment of toxins, e.g., *Perforafilum tunnelli*, produces seven of the 57 known toxins found in the Laguna Madre mats [66]. This assignment of species and toxicity is essential for understanding the impacts of toxin production and potential utility of these compounds as done for dolastatins and other cyanobacterial bioactive compounds [68].

## 5. Conclusions

In summary, the hypersaline mats found in the Laguna Madre potentially appear to differ from other systems based on dominance of multiple species, cryptic undescribed species, and different environmental drivers. The fate of these mats depends on the rate of sea level rise, and ability of these mats to migrate to high elevations as encroached by seawater. Future work will include seasonal analysis of these mats using these combined approaches, coupled with production and nitrogen fixation, to better understand the ecosystem services provided by these unique organisms.

## Figures and Tables

**Figure 1 biology-09-00183-f001:**
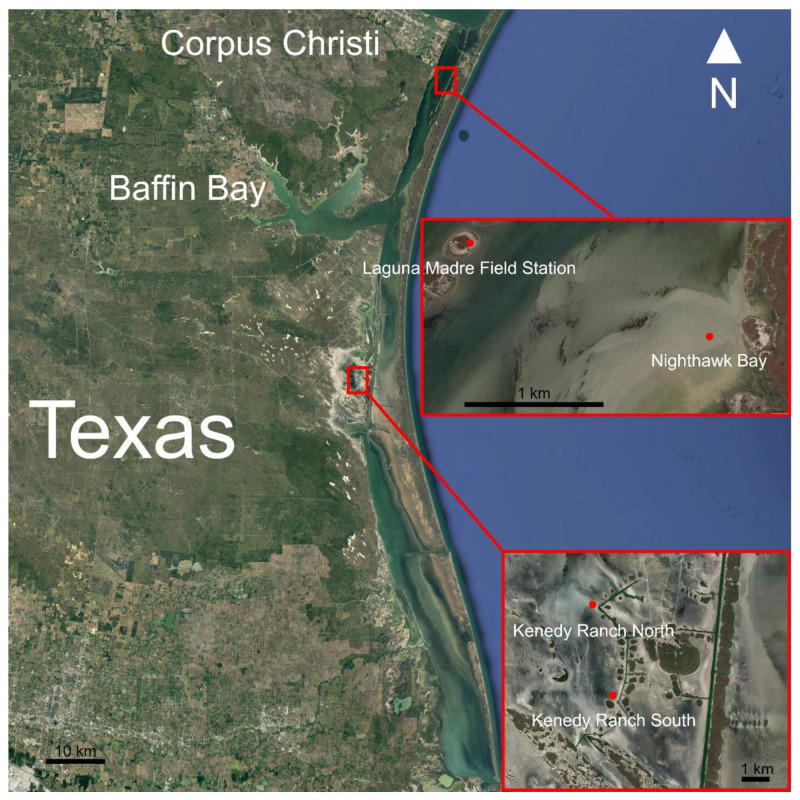
Map of the Laguna Madre showing the sample sites.

**Figure 2 biology-09-00183-f002:**
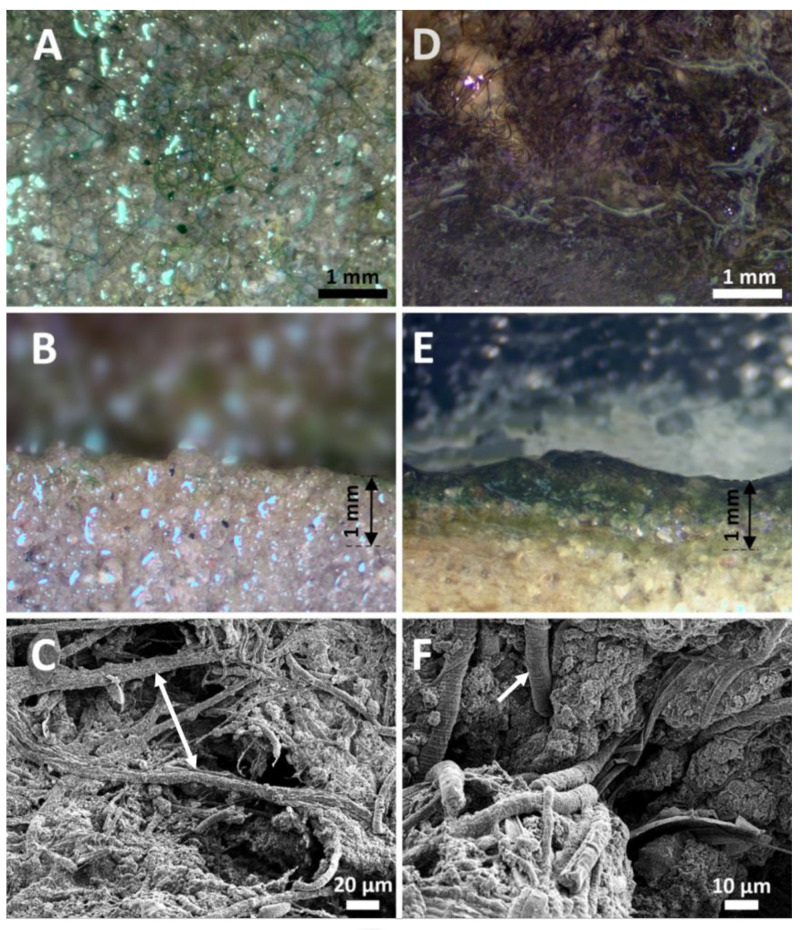
Light photographs and SEM images of submerged (**A**,**B**,**C**) and dry (**D**,**E**,**F**) mats. (**A**) Surface of a submerged, poorly developed mat with multitrichal filaments of the dominant species *Coleofasciculus chthonoplastes*. Note limited cyanobacterial coverage. (**B**) Cross section of the submerged mat with barely seen surface layer of cyanobacteria. Some filament occurred within the mat between the sand grains, at depths of ~0.2 mm due to active locomotion of *C*. *chthonoplastes*. (**C**) SEM image of the submerged mat with threads of *C*. *chthonoplastes* embracing mineral’s particles. (**D**) Surface of a dry, well developed mat with abundant filaments of the “*Lyngbya*” sp. covering almost 95% of area. (**E**) Cross section of a dry mat with visible colonies of “*Lyngbya*” sp. on the surface and blue-green layer underneath with other filamentous taxa. Note that the mat has pigmentation/coloration all the way down to 1 mm compared to thinner mats in submerged mat. (**F**) SEM picture of a dry mat with large rounded filaments of “*Lyngbya*” sp. having short discoid cells.

**Figure 3 biology-09-00183-f003:**
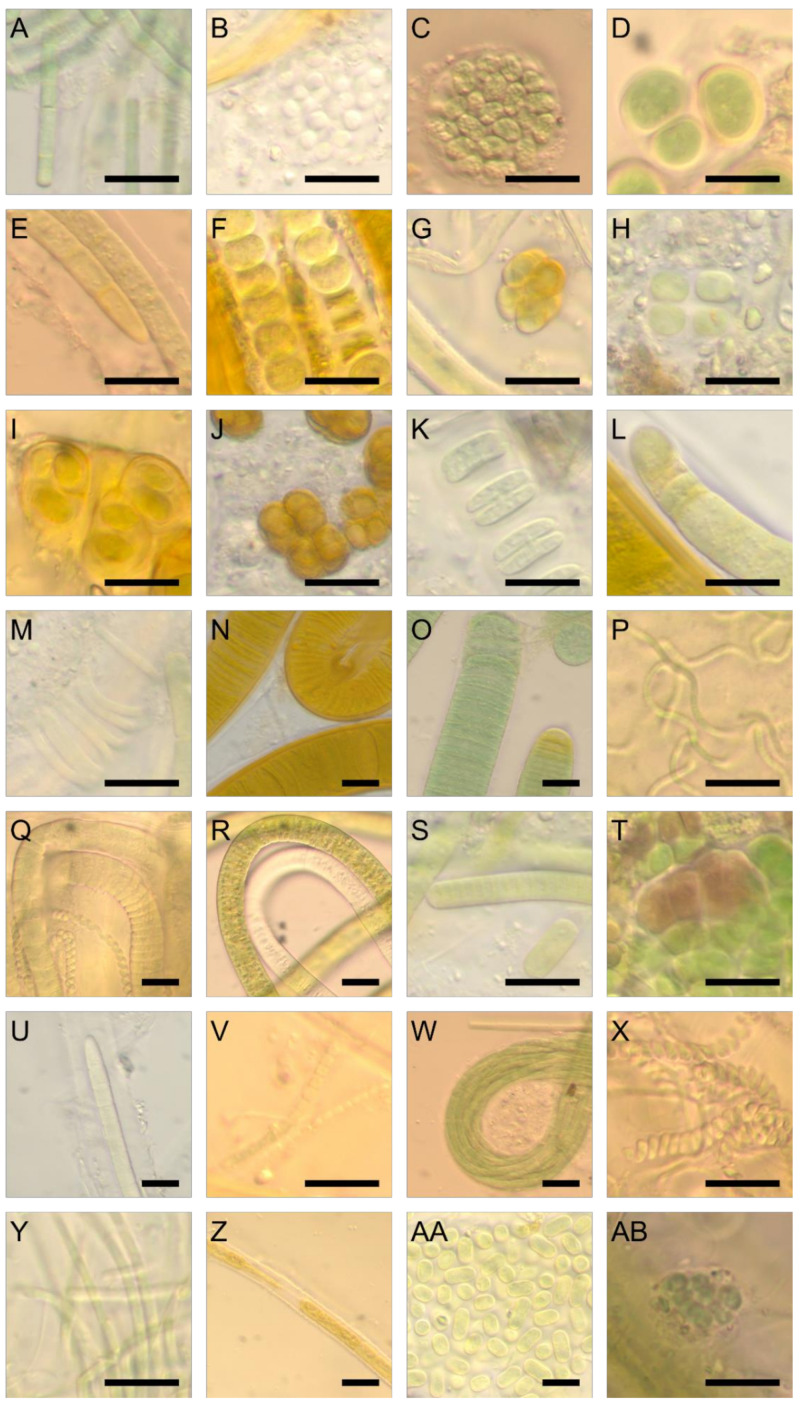
Representative morphologies of cyanobacterial taxa identified using light microscopy. (**A**) *Anagnostidinema* sp., a filamentous cyanobacterium with rounded end. (**B**) *Aphanocapsa* sp., a coccoid form with thick mucilage covering the colonies. (**C**) *Aphanothece* sp., a coccoid cyanobacterium with thin layer of mucilage. (**D**) *Chroococcus* sp., a coccoid form with thick mucilage covering each cell. (**E**) *Coleofasciculus chthonoplastes* (Thuret ex Gomont) M. Siegesmund et al., a filamentous form with tapered end and thick sheath covering the trichomes. (**F**) *Cyanocohniella* sp., a thermal cyanobacterium that was originally isolated from thermal springs. (**G**) *Cyanosarcina* sp., a unicellular-colonial, usually 2–16-multicelled cyanobacterium. (**H**) *Eucapsis prescottii* (Drouet et Daily) Komárek et Hindák, a unicellular-colonial, irregular, and diffluent mucilage-forming cyanobacterium. (**I**) *Gloeocapsa* sp., a unicellular-colonial cyanobacterium in a form of irregular aggregations with brown sheath. (**J**) *Gloeocapsopsis* sp., a less spherical unicellular-colonial cyanobacterium with brown sheath. (**K**) *Johannesbaptistia pellucida* (Dickie) W. R. Taylor et Drouet., a unicellular-pseudofilamentous taxon, that possessed a gelatinous sheath. (**L**) *Kamptonema* sp., an unsheathed cyanobacterium with rounded ends. (**M**) *Leibleinia* sp., a thin sheath-covered filamentous cyanobacterium that tangled in old stages. (**N**) *Lyngbya* sp., a filamentous cyanobacterium with brown sheath. (**O**) *Lyngbya* sp. with a thickened outer cell wall at the end cell. (**P**) *Nodosilinea* sp., a filamentous cyanobacterium with nitrogen-fixing potentials. (**Q**) *Oscillatoria nigro-viridis* Thwaites ex Gomont, a motile filamentous cyanobacterium. (**R**) *Perforafilum tunnelli*, a tapered-ended, sometimes hooked-ended, filamentous cyanobacterium with boring ability. (**S**) *Phormidium* sp., a filamentous cyanobacterium with tight sheath. (**T**) *Pleurocapsa* sp., a cyanobacterium that is unicellular-pseudofilamentous and crust-like, composed of irregular groups of cells. (**U**) *Porphyrosiphon* sp., a tapered-ended filamentous cyanobacterium with a firm and thick sheath. (**V**) *Pseudanabaena* sp., a filamentous cyanobacterium. (**W**) *Schizothrix* sp., a filamentous colonial cyanobacterium. (**X**) *Spirulina tenerrima* Kützing ex Gomont., a filamentous screw-like cyanobacterium. (**Y**) *Stenomitos* sp., a rounded-ended filamentous cyanobacterium. (**Z**) *Symplocastrum* sp., a filamentous cyanobacterium. (**AA**) *Synechococcus* sp., a unicellular-colonial, oval-shaped cyanobacterium without mucilage. (**AB**) *Xenococcus* sp., a unicellular-colonial cyanobacterium containing mucilaginous sheath. Scale bar in each figure represents 10 µm.

**Figure 4 biology-09-00183-f004:**
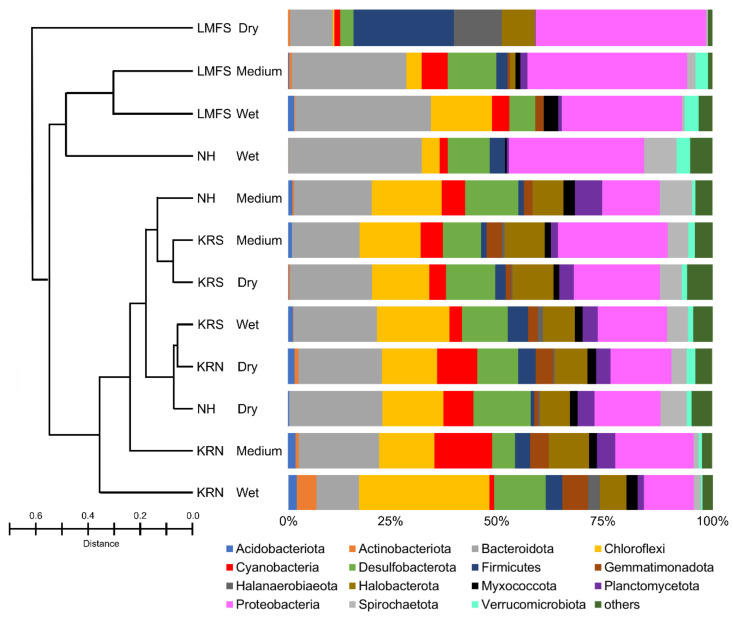
Bar plot showing the relative abundance (%) of amplified sequences of bacterial and archaeal phyla. Abundances were normalized to the total number of sequences, and taxonomic groups that were outside of the scope of research were grouped under the others category. The 15 most abundant orders across all samples are displayed in the legend. Bray–Curtis dissimilarity index is presented in the left panel.

**Figure 5 biology-09-00183-f005:**
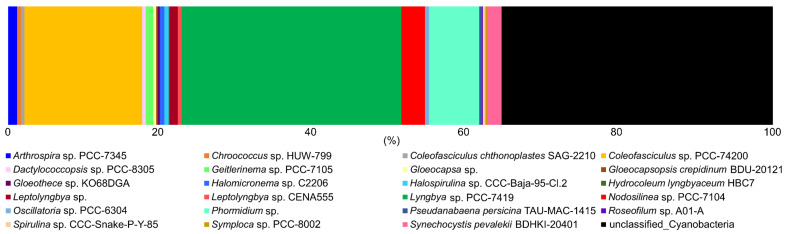
Bar plot showing the relative abundance (%) of amplified sequences of total cyanobacterial species in mid and upper Laguna Madre. Abundances were normalized to the total number of sequences.

**Figure 6 biology-09-00183-f006:**
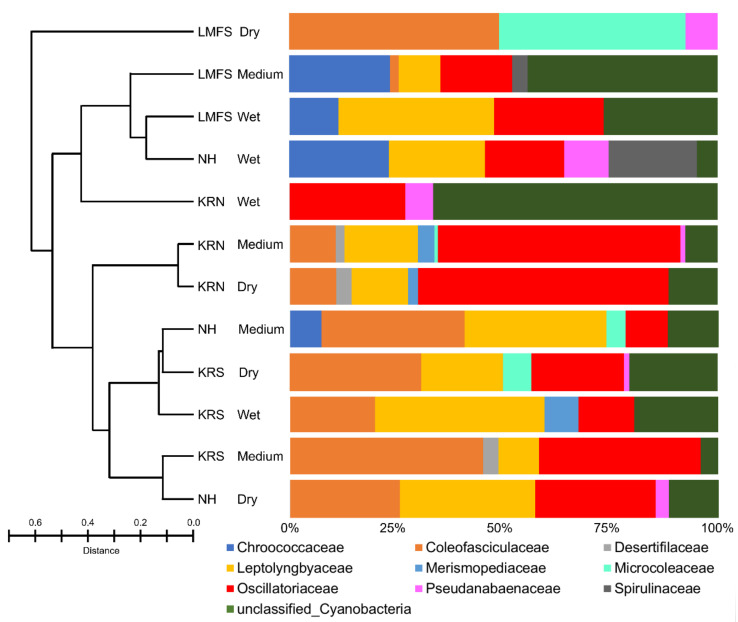
Bar plot showing the relative abundance of cyanobacterial amplified sequences separated by regions and conditions. Abundances were normalized to the total number of sequences. Bray–Curtis dissimilarity index is shown on the left.

**Table 1 biology-09-00183-t001:** Previous studies of microbial communities from hypersaline benthic communities.

					Dominant Taxa		
Environments	Salinity	Temp (°C)	Location	Analyses	Bacteria	Cyanobacteria	Reference
Brine mats	159–310	n.d.	Great Salt Plains, USA	Cloning, LM	n.d.	*Geitlerinema*, *Komvophoron*	[17]
	215	23	Eilat, Israel	LM, isolation	Desulfocella, Halochromatium, Methanohalobium	*Halothece*, *Halospirulina*, *Oscillatoria*	[18]
Estuarine mats	0–33.6	26.5–30.1	Pearl Estuary, China	Sanger	Bacteroidetes, Proteobacteria	Found, but not specified	[19]
	29–31.2	25	Tokyo Bay, Japan	Cloning	Proteobacteria	n.d.	[20]
Hypersaline lake mats	150	45	Solar Lake, Egypt	LM, SEM	n.d.	*Microcoleus* (syn. *Coleofasciculus*)	[21]
	69–108	17.3–25.5	Lake Clifton, Garden Lake, Government House Lake, Serpentine Lake, Herschell Lake, Australia	LM	n.d.	*Aphanothece*, *Oscillatoria*, *Microcoleus*, *Spirulina*, *Schizothrix*, *Gloeocapsa*	[22]
Lagoonal mats	27–36	0.1–22.5	Hooksiel, Germany	NGS, SEM, LM	n.d.	*Coleofasciculus*, *Hydrocoleum*, *Lyngbya*	[23]
	34.2–35.1	28.3–30.6	La Réunion lagoon, France	Cloning, LM	n.d.	*Anabaena*, *Hydrocoleum*, *Leptolyngbya*, *Lyngbya*, *Symploca*	[24]
	35–60	24–34	Araruama Lagoon, Brazil	Sanger, NGS, LM	n.d.	*Geitlerinema*, *Halomicronema*, *Microcoleus*, *Oxynema*	[25]
	14–32	30–37	Paje lagoon, Tanzania	Cloning, LM	Nitrogen-fixing bacteria detected	*Lyngbya*, *Microcoleus*, *Oscillatoria*, *Spirulina*	[26]
	78–90	n.d.	Guerrero Negro, Mexico	Cloning	Azotobacter, Desulfovibrio	*Anabaena*, *Halothece*, *Myxosarcina*, *Phormidium*, *Plectonema*, *Synechocystis*	[27]
	90	n.d.	Guerrero Negro, Mexico	Sanger	Bacteroidetes, Chloroflexi, Planctomycetes, Proteobacteria	*Lyngbya*	[28]
	225–280 (pore water)	n.d.	Schiermonnikoog Island, Netherlands	Sanger	n.d.	*Coleofasciculus*, *Lyngbya*	[29]
	0–150	10–32	Laguna Madre, USA	NGS, LM, SEM	Bacteroidetes, Chloroflexi, Desulfobacteria, Firmicutes, Halobacterota, Proteobacteria	*Coleofasciculus*, *Lyngbya*	This Study
Meltwater ponds	0.158–52.9 (mS/cm; cond.)	n.d.	McMurdo Ice Shelf, Bratina Island	Cloning, LM	n.d.	*Oscillatoria*	[30]
	0.6–40.8 (mS/cm; cond.)	n.d.	Bratina Island and the Miers Valley	Sanger	Actinobacteria, Bacteroidetes, Chloroflexi, Proteobacteria	*Anabaena*, *Microcoleus*, *Phormidesmis*	[31]
Rivermouth mats	0	9–29	Mulargia River, Italy	FISH	Proteobacteria	n.d.	[32]
Salt ponds	75–103	20–23	Salins-de-Giraud, France	CLSM, DGGE	n.d.	*Microcoleus* (syn. *Coleofasciculus*), *Halomicronema*	[33]
	87–114	28–40	San Salvador Island, Bahamas	Sanger	n.d.	*Leptolyngbya*, *Phormidium*	[34]

Temp.–Temperature; n.d.–not determined; LM–Light Microscopy; SEM–Scanning Electron Microscopy; NGS–NextGen Sequencing; FISH – Fluorescence in situ Hybridization; CLSM–Confocal Laser Scanning Microscopy; DGGE–Denaturing gradient gel electrophoresis.

**Table 2 biology-09-00183-t002:** List of the cyanobacterial species found in the sand flat mats in Laguna Madre. Med-medium; R-rare; C-common; A-abundant.

Order	Family	Genus/Species	Wet	Med	Dry	Genomic
Chroococcales	Aphanothecaceae	*Aphanothece utahensis* Tilden	R			
	Chroococcaceae	*Chroococcus* cf. *pulcherrimus* Welsh				
		*Chroococcus obliteratus* P. G. Richter	R			
		*Chroococcus* sp.	R	R	R	R
		*Cyanosarcina* sp.				
		*Dactylococcopsis* sp.				R
		*Gloeocapsa* sp.	R			R
		*Gloeocapsopsis* sp.		R	R	R
		*Gloeothece* sp.				R
	Cyanothrichaceae	*Johannesbaptistia pellucida* (Dickie) W. R. Taylor et Drouet	R	R	R	
		*Johannesbaptistia* sp.	R	R	R	
Chroococcidiopsidales	Chroococcidiopsidaceae	*Chroococcidiopsis* sp.				
Nostocales	Aphanizomenonaceae	*Cyanocohniella* sp.	R	R		
Oscillatoriales	Coleofasciculaceae	*Anagnostidinema* sp.	R		R	
		*Coleofasciculus chthonoplastes* (Thuret ex Gomont) M. Siegesmund et al.	R	R	R	R
		*Coleofasciculus* sp.	R	R	R	A
		*Roseofilum* sp.				R
	Gomontiellaceae	*Komvophoron* sp.	R			
	Microcoleaceae	*Arthrospira* sp.				C
		*Hydrocoleum lyngbyaceum*				R
		*Kamptonema* sp.	R		R	
		*Porphyrosiphon* sp.		R	R	
		*Symploca* sp.				R
		*Symplocastrum* sp.	R	R	R	
	Oscillatoriaceae	*Lyngbya* sp.	C	A	A	A
		New *genus* - akin to *Lyngbya aestuarii* Liebman ex Gomont	R	C	A	
		*Oscillatoria nigro*-*viridis* Thwaites ex Gomont	R			
		*Oscillatoria* sp.	C			R
		*Phormidium* sp.	R			C
Pleurocapsales	Hyellaceae	New genus - akin to *Pleurocapsa minuta* Geitler			R	
	Xenococcaceae	*Xenococcus* sp.	R			
Spirulinales	Spirulinaceae	*Halospirulina* sp.				R
		*Spirulina* sp.	C			R
		*Spirulina tenerrima* Kützing ex Gomont	C	C	C	
Synechococcales	Leptolyngbyaceae	*Halomicronema* sp.	R			R
		*Leibleinia nordgaardii* (Wille) Anagnostidis et Komárek	R			
		*Leibleinia* sp.	R	R	R	
		*Leibleinia subtilis* (Holden) Anagnostidis et Komárek	C			
		*Leptolyngbya* sp.				C
		*Nodosilinea* sp.				C
		*Stenomitos* sp.	R			
	Merismopediaceae	*Aphanocapsa* sp.	C		R	
		*Eucapsis prescottii* (Drouet et Daily) Komárek et Hindák	R		R	
		*Synechocystis* sp.	R			C
	Pseudanabaenaceae	*Geitlerinema* sp.	C			C
		*Jaaginema* sp.	R			
		*Pseudanabaena* sp.	C	C	C	R
	Schizotrichaceae	*Schizothrix helva* Frémy		R	R	
		*Schizothrix* sp.	R	R	R	
	Synechococcaceae	*Bacularia* sp.	C			
		*Synechococcus* sp.		C	R	
	Trichocoleusaceae	*Trichocoleus* sp.			R	
		*Trichocoleus tenerrimus* (Gomont) Anagnostidis			R	
Oscillatoriales	Laspinemaceae fam. nov.	*Perforafilum tunnelli* (new genus - akin to *Oxynema* sp.)	R			

**Table 3 biology-09-00183-t003:** HPLC Orbitrap results for Laguna Madre wind-tidal flat screening for all known cyanobacterial bioactive metabolites, organized as in Huang and Zimba (2019). Listed compounds had peak exceeding 10,000 area counts. Names followed by an asterisk were area counts exceeding 100,000, with a double asterisk indicating mass difference equals 0 ppm. Occurrence (Occr) refers to how many samples (of 9 total) contained each toxin. Literature of metabolite origins was summarized in Huang and Zimba (2019). Occr—bioactive metabolite occurrence; Cy—Cyanobacteria; Y—yes; N—no; N/A—not applicable.

					Literature Origins	
Main Group	Subgroup	Compound	Bioactivity	Occr	Cy	Family	Genus	In This Study
Aeruginosin		AER 103-A	Protease inhibitors	1	Y	Microcystaceae	*Microcystis*	Order found
AER 98-A	2				
AER 98-B	1				
AER 98-C	2				
AER DA 642-A	1				
AER DA 642-B	1				
AER DA-688	3				
AER GE-686	1				
AER KB-676	1				
Spumigin	AER-SPM B-1	1	Y	Nostocaceae	*Nodularia*	Morph found
AER-SPM B-2*	2				
AER-SPM E*	1				
AER-SPM G	1				
AER-SPM H	1				
Anabaenopeptin		ABPN B	Protease/protein phosphate inhibitors	5	Y	Pseudanabaenaceae	*Toxifilum*	Family found
Feritonic acid	ABPN-FTA B	1	Y	Microcystaceae	*Microcystis*	Order found
Konbamide	ABPN-KBA	1	N	Theonellidae	*Theonella*	N/A
Paltolide	ABPN-PAL A	2	N	Theonellidae	*Theonella*	N/A
Anatoxin		ANA-a	Neurotoxicity	4	Y	Microcoleaceae	*Arthrospira*	Genus found
11-carboxy-ANA	1				
Antillatoxin		ATX B	Cytotoxicity	1	Y	Oscillatoriaceae	*Moorea*	Family found
Apratoxin		APX F	Cytotoxicity	1	Y	Oscillatoriaceae	*Moorea*	Family found
APX G	1				
Coibacin		CBC C	Cytotoxicity, anti-parasite, neurotoxicity	4	Y	Oscillatoriaceae	*Oscillatoria*	Genus found
CBC D	1				
Cyanopeptolin	Aeruginopeptin	CPT-APP 228-A	Protease/protein phosphate inhibitors, cytotoxicity, anti-crustacean	1	Y	Microcystaceae	*Microcystis*	Order found
CPT-APP 917-S-C	1				
Micropeptin	CPT-MPP MM-836	1	Y	Microcystaceae	*Microcystis*	Order found
Oscillapeptin	CPT-OPP A	2	Y	Microcoleaceae	*Planktothrix*	Family found
Scyptolin	CPT-SPT A	1	Y	Scytonemataceae	*Scytonema*	Morph found
Tasipeptin	CPT-TSP B	1	Y	Microcoleaceae	*Symploca*	Genus found
Largamide	Dolastatin	LRA-DST D	Neurotoxicity, cytotoxicity	2	N	Aplysiidae	*Dolabella*	N/A
Linear-Dolastatin		l-DST 15	Protease inhibitors, cytotoxicity, anti-bacteria, anti-fungi, anti-parasite	1	Y	Microcoleaceae	*Symploca*	Genus found
l-DST 18	1				
Lyngbyabellin		LBB F	Cytotoxicity, anti-fungi	1	Y	Oscillatoriaceae	*Moorea*	Family found
LBB G	1				
Lyngbyaloside	Phorbaside	LBL-PBA A*	Cytotoxicity	2	Y	Oscillatoriaceae	*Moorea*	Family found
Malyngamide		MGA 4	Cytotoxicity, anti-fungi, anti-parasite	2	Y	Oscillatoriaceae	*Moorea*	Family found
Microcolin	Majusculamide	MCL-MSA D	Cytotoxicity	1	Y	Oscillatoriaceae	*Moorea*	Family found
Microcyclamide		MCA 7806-B	Cytotoxicity, anti-parasite	7	Y	Microcystaceae	*Microcystis*	Order found
Aerucyclamide	MCA-ACA C	2	Y	Microcystaceae	*Microcystis*	Order found
Balgacyclamide	MCA-BCA B*	1	Y	Microcystaceae	*Microcystis*	Order found
	MCA-BCA C	5				
Bistratamide	MCA-BTA I	5	N	Didemnidae	*Lissoclinum*	N/A
Venturamide	MCA-tetraOH-VTA A	1	Y	Oscillatoriaceae	*Oscillatoria*	Genus found
Microcystin		[(6Z)-Adda5]MCYST-RR	Hepatotoxicity	1	Y	Microcystaceae	*Microcystis*	Order found
[ADMAdda5]MCYST-HilR	4				
[ADMAdda5]MCYST-LHar	4				
[ADMAdda5]MCYST-LR	4				
[D-Asp3, ADMAdda5]MCYST-LHar	3				
[D-Asp3, Dha7]MCYST-E(OMe)E(OMe)	7				
MCYST-OiaA	1				
MCYST-OiaAba	3				
MCYST-OiaR	1				
MCYST-RF	1

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
