# Peer review of "Preliminary Assessment of Microbial Community Structure of Wind-Tidal Flats in the Laguna Madre, Texas, USA"

_biology, 2020, doi:10.3390/biology9080183_

Round 1
Reviewer 1 Report
I think the data is well presented and written well enough. I would suggest making it clearer to the reader what the relevance/of the study is (why should people care). What does it mean to people that the microbial community is different in Laguna Madre than other places? Does it likely mean impacts to /animal health, etc.?
Author Response
Author response: This has been addressed in the first paragraph of the Discussion. Also added “The drilling process of harvesting these natural sources may have resulted in the richness of sulfur-containing substances, which support the success of species in the Desulfobacteria. Alternatively, the seasonal deposition of seagrass wrack provides an anaerobic habitat suitable for growth of these taxa.” in the second paragraph.
Reviewer 2 Report
Review of biology-866877: Preliminary assessment of microbial community structure of wind-tidal algal flats in the Laguna Madre, Texas, USA for Biology
The authors assessed the cyanobacterial and bacterial structure of wind-tidal flats in the Laguna Madre in the USA using a polyphasic approach. Moreover, the authors examined mats in terms of numerous bioactive compounds. They revealed high biodiversity of cyanobacteria and detected several groups of bacteria in the mats. In my opinion, the topic and the results presented by the authors are very interesting and expand the knowledge about microbes forming and living in mats in such extreme habitat. Noteworthy is that the authors used modern polyphasic approach in their work and included photos of cyanobacteria found in the samples. The manuscript is well written and structured. Graphs and photographic documentation were prepared reliably and have good quality. I think the manuscript might be interesting for a broad audience. My overall recommendation is ‘minor revision’.
My detail comments and suggestions are presented below.
Title:
I suggest removing the word ‘algal’ from the title as the authors were focused in their work on cyanobacteria and bacteria and these organisms are not algae.
Introduction
Line 42: Please inverse citations (from older to newer). Please check it in the whole manuscript.
Lines 50-51: Please provide a reference for avian residents of the Laguna Madre.
Lines 60-61: Please provide some most dominant species names of diatoms.
Line 71: What is the rationale for including the table in the introduction. I suggest presenting it in the supplementary materials.
In the last paragraph, the authors should explain why their research is important and needed. Moreover, the research problem should also be highlighted. Currently, only research aim is provided.
Materials and Methods
Line 106: Please provide the name of the country. The same: lines 134, 138
Line 117: The state Indiana should be in the form of abbreviation.
Results
Line 164-165: Please consider modifying the beginning of the sentence: "Species of several cyanobacteria genera (Geitlerinema,...) were...".
Figure 2: Please consider adding an arrow to help inexperienced reader finding the objects (panel C and F). Please replace the term ‘colonies’ by filament aggregations (E).
Table 2: Please provide the name of order and family name for the new genus found in the mats (Perforafilum tunnelli)
Line 273: Oscillatoriaceae
Table 3: Please adjust the width of columns in the table to improve the visibility of subgroups of bioactive metabolites and Family/Genus names
Discussion
I have no comments on this section. The authors discussed the obtained results in various aspects and in relation to appropriate literature.
Author Response
Title:
I suggest removing the word ‘algal’ from the title as the authors were focused in their work on cyanobacteria and bacteria and these organisms are not algae.
Author response: Addressed.
Introduction
Line 42: Please inverse citations (from older to newer). Please check it in the whole manuscript.
Author response: Addressed.
Lines 50-51: Please provide a reference for avian residents of the Laguna Madre.
Author response: Addressed.
Lines 60-61: Please provide some most dominant species names of diatoms.
Author response: Addressed. Added Amphora normanii Rabenhorst, Amphora spp., Nitzschia spp.
Line 71: What is the rationale for including the table in the introduction. I suggest presenting it in the supplementary materials.
Author response: Our rationale was to give readers a more organized understanding of the comparison between our study and published literature. Besides, moving this table to SM reduces the utility and visibility of this summary table.
In the last paragraph, the authors should explain why their research is important and needed. Moreover, the research problem should also be highlighted. Currently, only research aim is provided.
Author response: Added “There is an urgency to understand unique habitats such as these hypersaline mats, as sea-level rise and Anthropocene impacts are likely to alter their habitat. The isolation and identification of these organisms may provide unique resources for natural product discovery.”
Materials and Methods
Line 106: Please provide the name of the country. The same: lines 134, 138
Author response: Addressed.
Line 117: The state Indiana should be in the form of abbreviation.
Author response: Addressed.
Results
Line 164-165: Please consider modifying the beginning of the sentence: "Species of several cyanobacteria genera (Geitlerinema,...) were...".
Author response: Addressed.
Figure 2: Please consider adding an arrow to help inexperienced reader finding the objects (panel C and F). Please replace the term ‘colonies’ by filament aggregations (E).
Author response: Addressed.
Table 2: Please provide the name of order and family name for the new genus found in the mats (Perforafilum tunnelli)
Author response: Addressed.
Line 273: Oscillatoriaceae
Author response: Addressed.
Table 3: Please adjust the width of columns in the table to improve the visibility of subgroups of bioactive metabolites and Family/Genus names
Author response: We think this has to be done by the designing team of MDPI.
Discussion
I have no comments on this section. The authors discussed the obtained results in various aspects and in relation to appropriate literature.